# Effects of supplemental irrigation at the jointing stage on population dynamics, grain yield, and water-use efficiency of two different spike-type wheat cultivars

**Yunqiu Shang**[1 º], **Xiang Lin**[1], **Ping Li**[1º], **Shubo Gu**[1], **Keyi Lei**[1], **Sen Wang**[1], **Xinhui Hu**[1], **Panpan Zhao**[1], **Dong Wang**[1,2]\*

**1** College of Agronomy, Shandong Agricultural University, State Key Laboratory of Crop Biology, Key Laboratory of Crop Ecophysiology and Farming System, Ministry of Agriculture, Taian, Shandong, People's Republic of China, **2** Zibo HeFeng Seed Technology co., ltd., Linzi, Zibo, Shandong, People's Republic of China

º These authors contributed equally to this work.

\* wangd@sdau.edu.cn

## Abstract

To solve the problems of yield reduction and low water-use efficiency (WUE) of winter wheat (*Triticum aestivum* L.) caused by winter and spring drought, a 2-year field experiment (2017–2019) was performed under movable shelter conditions with the large- and multispike cultivars Shannong 23 and 29, respectively, to explore the optimal supplemental irrigation regime. Three wetting layers were used for irrigation at the jointing stage: 0–10 cm (T2), 0–20 cm (T3) and 0–30 cm (T4). No irrigation at the jointing stage (T1) served as the control. Within a given cultivar, the soil water content in the 0–80 cm soil layers increased after irrigation, and the rate of tiller mortality decreased with increasing depth of the wetting layer used for irrigation at jointing. No significant differences were found between the T3 and T4 treatments in the photosynthetic rate (Pn) of the apical leaf of the main stem (O), the first primary tiller (I) and the fourth tiller (IV) after jointing. However, compared with the T3 treatment, the T4 treatment had a significantly higher transpiration rate (Tr) and lower instantaneous water-use efficiency ($WUE_{leaf}$) of the apical leaf of the O and tillers I and IV. This eventually led to a decreasing WUE, although there was no significant change in the spike number or grain yield. These results indicated that moderate irrigation at jointing can effectively reduce the tiller mortality, improve the leaf Pn of the tillers, and increase the spike number and grain yield. However, excessive irrigation can significantly increase the leaf Tr of the tillers, lead to inefficient water consumption and significantly reduce the $WUE_{leaf}$ of the tillers and the WUE. Irrigation at the jointing stage brought the soil water content in the 0–20 cm profile to 100% of field capacity, making it the most suitable supplemental irrigation regime for both the large- and multispike cultivars in the North China Plain.

**Data Availability Statement:** All relevant data are within the paper and its Supporting Information files.

**Funding:** This work was financially supported by the Major Scientific and Technological Innovation Project of Shandong Province (2019JZZY010716), the Taishan Industry Leader Talent Project of Shandong, the National Special Support Program for High-level Talents, and the Special Fund for Agro-scientific Research in the Public Interest (201503130). The author DW is part-time employed by and receives salary from Zibo HeFeng Seed Technology. The specific roles of these authors are articulated in the 'author contributions' section. The funders had no role in study design, data collection and analysis, decision to publish, or preparation of the manuscript.

**Competing interests:** The author DW is part-time employed by and receives salary from Zibo HeFeng Seed Technology. This does not alter our adherence to PLOS ONE policies on sharing data and materials. There are no patents, products in development or marketed products associated with this research to declare. The other authors declare no other competing interests exist.

## Introduction

In China, winter wheat is sown on $2.45 \times 10^6$ ha and yields $1.34 \times 10^8$ t grain [1], and more than 70% of the wheat yield is produced in the North China Plain [2]. However, water is the most important limiting factor for wheat production in this region [3]. The distribution of precipitation is uneven, with 70% of rainfall occurring from June to September, and water stress occurs regularly during the winter wheat growing period [4]. Since the 1960s, winter and spring precipitation have decreased, the frequency of water stress during the winter wheat growing period in North China has increased [5], and drought has caused more than a 60% loss of grain production [6,7]. Hence, irrigation plays a crucial role in achieving high grain yield for wheat. Groundwater is the main irrigation source, and more than 80% of the groundwater is used to irrigate winter wheat [3]. However, excessive water supply induces groundwater waste with decreasing water-use efficiency (WUE) [8]. Therefore, optimal irrigation regimes at the critical growth stage are imperative for improving WUE while maintaining a stable grain yield.

Water supply at the jointing stage is critical for yield formation of winter wheat, and drought in this period severely influences spike numbers and photosynthesis rates and reduces production [9,10,11]. Tillering ability is an important agronomic trait of wheat and an adaptive mechanism that allows the plants to obtain additional source organs [12] and determines grain yield by affecting the spike number per unit area [13,14]. Under rain-fed conditions, a significant positive correlation is found between the increase in grain yield and the number of spikes [15]. After irrigation, the absorbable moisture is mainly stored in the 0–60 cm soil layers, where the higher soil moisture condition is accompanied by higher tiller numbers [16]. The increased water supply is conducive to increasing productive tiller number and improving grain yield [17]. Therefore, it is necessary to investigate the changes in population dynamics after irrigation at jointing to determine a suitable irrigation scheme for constructing high-yield and high-efficiency populations. However, different wheat cultivars have different tillering abilities, spike formation rates and spike numbers. The spike number per plant is 1.36–1.82 stems in the large-spike cultivars and 3.37–3.85 stems in the multispike cultivars [18]. The low number of tillers and the low rate of productive tillers are important factors in restricting the yield, and these factors limit the development of the productive potential in large-spike cultivars to an extreme extent [19]. The spike number of a large-spike cultivar was significantly decreased, but the grain yield increased by 9.8–16.5% compared with that of a multispike cultivar under the same growth environment [20]. However, Gaju et al. (2009) reported that the spike number, grain number and grain yield of large-spike cultivars decreased by 26%, 23% and 8%, respectively, compared with those of multi-spike cultivars [21]. Therefore, it is important to study the dynamic change in the tiller number and spike number of different spike-type cultivars after irrigation at the jointing stage to implement effective water control specific to cultivars and realize high yield and high efficiency.

The jointing stage is the period with dramatic and dynamic tiller changes. Irrigation at the jointing stage yields a positive correlation between the formation of productive tillers and the increase in grain yield. However, the relation of the soil moisture condition at the jointing stage and the spike formation of tillers is rarely reported. The objectives of this study were (1) to determine the relation of different wetting layers for irrigation at the jointing stage with soil moisture content and population dynamics of two cultivars, (2) to clarify the response of the gas exchange characteristics of the apical leaves of different tillers to irrigation at the jointing stage, and (3) to clarify the influence of different wetting layers for irrigation at the jointing stage on grain yield and WUE.

## Materials and methods

### Experimental site

The field experiment was performed in Daolang, Shandong Province, China (36°12′ N, 116° 54′ E), in two growing seasons (from October 2017 to June 2018 and from October 2018 to June 2019). This area has a warm temperate semi-humid continental monsoon climate with an annual average temperature of 13.6 °C and an annual average precipitation of 621.2 mm. Precipitation is concentrated in the summer, and approximately 40% of the precipitation falls during the winter wheat growing season. The groundwater depth is 25 m. The soil is silty loam, and corn (*Zea mays* L.) was the previous crop. The organic matter, total nitrogen, hydrolysable nitrogen, available phosphorous, and available potassium in the 0–20 cm soil layer of the experimental field before sowing are listed in Table 1. The bulk density, field capacity, and relative water content in the 0–200 cm soil layers of the experimental field before sowing are listed in Table 2. The seasonal precipitation in 2017–2018 and 2018–2019 is shown in Table 3.

### Experimental design

Three wetting layers were used for irrigation at the jointing stage: 0–10 cm (T2), 0–20 cm (T3), and 0–30 cm (T4). A treatment with no irrigation at the jointing stage (T1) served as the control. The amount of supplemental irrigation (SI) was calculated by $I = 10 \cdot \gamma \cdot H \cdot (FC-\beta_j)$ [22], where I (mm) is the amount of SI, $\gamma$ (g·cm$^3$) is the soil bulk density, H (cm) is the soil depth of the wetting layer, FC (%) is the field capacity, and $\beta_j$ (%) is the soil water content before irrigation. All the treatments were performed under movable shelter conditions, and the canopy was covered when rain fell during the period from wintering to anthesis.

The amount of SI from sowing to wintering and from anthesis to maturity was determined by a method of on-demand SI for winter wheat [23]. The amount of SI was based on the soil water storage at sowing and precipitation in various growth stages. The two-year field water management at sowing was conducted as follows: the soil water storage in the 0–100 cm soil layer ($S_s$, mm) was predicted by using the 0–40 cm soil water content ($\theta_{v-0-40}$, %) at sowing, which is feasible when $S_s = 7.265\theta_{v-0-40}+100.68$. When the soil relative water content ($\theta_{v-0-20}$, %) in the 0–20 cm soil layer at sowing is greater than 70%, no water is applied; at less than 70%, the formula $I_s = 10 \times 0.02 \times (FC_{0-20}-\theta_{v-0-20})$ is used to calculate the amount of irrigation. $FC_{0-20}$ (v/v, %) and $\theta_{v-0-20}$ (v/v, %) are the field capacity and the soil water content in the 0–20 cm soil layer, respectively. The two-year field water management at wintering was as follows: when the main water supply amount, calculated as $WS_{sw} = S_s+P_{sw}+I_s$, is more than 326.8 mm during the period from sowing to wintering, SI is not required; when $WS_{sw}$ is less than 326.8 mm, the equation $I_w = 326.8-WS_{sw}$, where $WS_{sw}$ (mm) is the main water supply during the period from sowing to wintering and $P_{sw}$ (mm) is the precipitation amount during the period from sowing to wintering, is used to calculate the amount of irrigation at wintering ($I_w$, mm). The 2-year field water management at anthesis was as follows: the amount of SI at anthesis ($I_a$) is determined by $I_a = SI_{sa}-I_w-I_j$, where $SI_{sa}$ (mm) is the amount of SI from seeding to anthesis and is calculated by $SI_{sa} = -0.022\ Y_s+224.742$ and $I_j$ (mm) is the amount of SI at jointing. $Y_s = $

**Table 1. Soil nutrient contents in the 0–20 cm soil layer of the experimental field before sowing.**

| Year | Organic matter (g·kg$^{-1}$) | Total nitrogen (g·kg$^{-1}$) | Hydrolysable nitrogen (mg·kg$^{-1}$) | Available phosphorus (mg·kg$^{-1}$) | Available potassium s(mg·kg$^{-1}$) |
|---|---|---|---|---|---|
| 2017–2018 | 1.15 | 1.29 | 93.09 | 45.49 | 142.62 |
| 2018–2019 | 0.97 | 1.05 | 82.60 | 55.55 | 146.95 |

**Table 2. Soil bulk density, field capacity, and soil relative water content in the 0–200 cm soil layer of the experimental field before sowing.**

| Soil layer (cm) | Soil bulk density (g·cm$^{-3}$) | Field capacity (%) | Relative soil water content before sowing (%) | |
|---|---|---|---|---|
| | | | 2017–2018 | 2018–2019 |
| 0–20 | 1.41 | 28.9 | 72.60 | 55.13 |
| 20–40 | 1.60 | 23.1 | 58.73 | 77.09 |
| 40–60 | 1.39 | 27.4 | 60.11 | 69.61 |
| 60–80 | 1.49 | 27.4 | 67.71 | 73.69 |
| 80–100 | 1.54 | 26.0 | 68.20 | 82.83 |
| 100–120 | 1.60 | 24.5 | 72.60 | 82.00 |
| 120–140 | 1.62 | 23.9 | 80.91 | 85.00 |
| 140–160 | 1.62 | 23.5 | 82.62 | 87.97 |
| 160–180 | 1.61 | 23.7 | 83.84 | 89.30 |
| 180–200 | 1.62 | 24.0 | 85.24 | 90.02 |

$35.776\,S_{si}+6.831\,P_{sw}+10.103\,P_{wj}+10.064\,P_{ja}+9.476\,P_{am}-5250.452$ ($Y_s$, kg·ha$^{-1}$) is used to predict the grain yield of winter wheat under rain-fed conditions. $P_{wj}$ (mm) is the precipitation amount during the period from wintering to jointing, $P_{ja}$ (mm) is the precipitation amount during the period from jointing to anthesis and $P_{am}$ is 0. The amount of SI at each growth stage under the different treatments is shown in Table 4.

## Crop management

Two typical cultivars of winter wheat with different spike types were used in this study. SN23 and SN29 were the large- and multispike cultivars with low and high tillers, respectively. Seeds were sown at 240 plants·m$^{-2}$ of SN23 and at 165 plants·m$^{-2}$ of SN29 on 2017–10–13 and at 300 plants·m$^{-2}$ of SN23 and at 225 plants·m$^{-2}$ of SN29 on 2018–10–8. The harvest data were 2018–6–8 and 2019–6–12. The compound fertilizer that contained N:P:K at 15:15:15 was applied at 900 kg ha$^{-1}$ as base fertilizer before sowing. The urea (N 46%) was top-dressed at 228 kg ha$^{-1}$ at the jointing stage.

## Measurements

**Soil water content and water consumption by winter wheat.** The 200 cm deep soil profiles used for the soil water content measurement samples ($n = 3$) were collected at 20 cm intervals by using a soil corer in each experimental plot. The measurements were executed before sowing or maturity, on the day before irrigation, and on the third day after irrigation. The soil water content was measured using an oven-drying method [24].

$$\text{Soil water content}\,(\%) = (\text{wet soil weight} - \text{dry soil weight})/\text{dry soil weight} \times 100\%$$

$$\text{Soil relative water content}\,(\%) = \text{soil water content}/\text{field capacity} \times 100\%$$

**Table 3. Precipitation amount (mm) in different growth stages of winter wheat.**

| Year | Sowing-wintering | Sheltering from the rain | Anthesis-maturity |
|---|---|---|---|
| | | Wintering-anthesis | |
| 2017–2018 | 4 | 0 | 121.2 |
| 2018–2019 | 28.8 | 0 | 45.2 |

**Table 4. Irrigation amount (mm) in each growth stage from 2017–2019.**

| Year | Cultivar | Treatment | Sowing | Wintering | Jointing | Anthesis | Total |
|------|----------|-----------|--------|-----------|----------|----------|-------|
| 2017–2018 | SN23 | T1 | — | 46.8 | — | — | 46.8 |
|  |  | T2 | — | 46.8 | 27.8 | — | 74.6 |
|  |  | T3 | — | 46.8 | 52.6 | — | 99.4 |
|  |  | T4 | — | 46.8 | 72.2 | — | 119.0 |
|  | SN29 | T1 | — | 46.8 | — | — | 46.8 |
|  |  | T2 | — | 46.8 | 27.8 | — | 74.6 |
|  |  | T3 | — | 46.8 | 52.6 | — | 99.4 |
|  |  | T4 | — | 46.8 | 72.2 | — | 119.0 |
| 2018–2019 | SN23 | T1 | 35.2 | — | — | 54.1 | 89.3 |
|  |  | T2 | 35.2 | — | 29.0 | 54.1 | 118.3 |
|  |  | T3 | 35.2 | — | 55.9 | 54.1 | 145.3 |
|  |  | T4 | 35.2 | — | 76.6 | 54.1 | 165.9 |
|  | SN29 | T1 | 35.2 | — | — | 54.1 | 89.3 |
|  |  | T2 | 35.2 | — | 29.0 | 54.1 | 118.3 |
|  |  | T3 | 35.2 | — | 55.9 | 54.1 | 145.3 |
|  |  | T4 | 35.2 | — | 76.6 | 54.1 | 165.9 |

SN23, SN29, and "—" indicate Shannong 23 and 29 and no irrigation, respectively.

The total water consumption of winter wheat was calculated using the soil water balance equation [25]: ET = ΔS + I + P − R − D + CR, where ET (mm) is the total water consumption, ΔS (mm) is the water consumption from the 0–200 cm soil layer during the growing season, I (mm) and P (mm) are the irrigation and precipitation, respectively, R (mm) is the runoff, D (mm) is the drainage from the root zone, and CR (mm) is the capillary rise to the root zone. Three factors in this equation (R, D and CR) can be ignored in this experimental site.

**Population dynamics of winter wheat.** Tillers were marked from the appearance of the first tiller in wheat. The newly initiated tillers of each plant were checked and tagged every 5 days. After jointing, the tagged plants were selected and separated according to the tiller positions for measuring. In this paper, the main stem is represented by O, and the primary tillers growing from the true leaf axillary of O are represented by I, II, III, IV, etc. Conversely, I-p, I-1, I-2, etc. are used for the secondary tillers growing from the axillary of the true leaf of the primary tillers [26].

The number of tillers (stems) per square meter was investigated at wintering, re-greening, standing, jointing, anthesis and maturity. At jointing, the number of tillers (stems) was investigated once every 7 days from 0 days after jointing to anthesis.

**Net Pn, transpiration rates (Tr) and instantaneous WUE (WUE$_{leaf}$).** The net Pn and transpiration rates (Tr) of the apical leaves of different tillers were measured at 0, 7, 14, 21, and 28 days after jointing by using a Li-6400 portable photosynthesis system (LI-Cor, Inc., Lincoln, Nebraska, USA) from 9:00–11:00 a.m. with an artificial light source. Five plants with marked tillers were selected for the measurement in each plot. The WUE$_{leaf}$ was calculated as the ratio of Pn to Tr [27].

**Grain yield and components.** At maturity, 1.5 m$^2$ of grain yield (kg·ha$^{-1}$) was harvested from each test plot and reported on a 12.5% wet basis through natural air drying. The 1000-grain weight was investigated by taking the average of six samples of 1000 grains. The grain number per spike was determined from fifteen wheat ears.

**WUE.** The WUE was calculated as the ratio of Y to ET, where WUE (kg·ha$^{-1}$·mm$^{-1}$) was the water-use efficiency for the grain yield, $Y$ (kg·ha$^{-1}$) was the grain yield, and ET (mm) was the water consumption amount during the growing season.

## Statistical analysis

Data related to the grain yield, stem number, grain per spike, 1000-grain weight, ET, and WUE were collated and analyses were calculated using SPSS (Statistical Product and Service Solutions, IBM, Armonk, New York, USA), and the data are shown as averages. The least significant difference (LSD) method (performed at a probability level of P ≤ 0.05) was used to determine the significant differences among the different treatments. Mapping was performed by SigmaPlot 12.5 (Systat Software Inc., Dundas, CA, GER) software.

## Results

### Soil relative water content after irrigation

After irrigation at the jointing stage, the relative water content in the 0–80 cm soil layer under the T2, T3, and T4 treatments increased by 16.2%, 18.3%, and 24.5% in SN23, and by 20.5%, 20.8% and 21.6% in SN29, compared with that under the T1 treatment (Fig 1). However, the relative water content in the 80–200 cm soil layer did not vary under different treatments in either cultivar. From anthesis to maturity, no significant differences were found in the relative water content in the 0–200 cm soil layers between the T3 and T4 treatments, but the relative water contents in the T3 and T4 treatments were greater than those in the T2 treatment.

### Water consumption at different growth stages

The water consumption of winter wheat during each growth stage in different treatments is listed in Table 5. The stage-dependent water consumption, which was affected by the wetting layer at jointing, showed a similar pattern between the cultivars. Throughout the whole growth season, the water consumption from sowing to wintering was the lowest. With an increase in the depth of the wetting layer, the water consumption from jointing to anthesis increased gradually, and the T4 treatment had significantly higher water consumption than that of the other treatments. A slight difference in water consumption was observed between the treatments from anthesis to maturity. The highest total water consumption of winter wheat was found in the T4 treatment, and no significant difference was found between the T2 and T3 treatments.

### Dynamics of the wheat population

The population dynamics in each wheat growth stage are shown in Fig 2. For the large-spike cultivar (SN23), the tiller numbers in the T2, T3, and T4 treatments from jointing to anthesis increased by 82, 172, and 191 stem·m$^{-2}$, respectively, and the average rate of decrease in tiller number was 3.15, 5.44, and 4.60 stem·m$^{-2}$·d$^{-1}$, respectively, compared with those in the T1treatment. For the multispike cultivar (SN29), the tiller numbers of the T2, T3, and T4 treatments from jointing to anthesis increased by 105, 113, and 143 stem·m$^{-2}$, respectively, and the average rate of decrease in tiller number was 2.34, 6.72, and 5.61 stem·m$^{-2}$·d$^{-1}$, respectively, compared with those of T1. These results indicated that the tiller death rate decreased with increasing depth of the wetting layer for irrigation at the jointing stage. The number of tillers in SN23 decreased rapidly under low soil moisture conditions but slowly under high moisture conditions compared with that of SN29.

The spike number at maturity in the T2, T3, and T4 treatments increased by 17.64%, 23.57%, and 24.84% in SN23 and by 6.57%, 11.55% and 14.08% in SN29 compared with that in

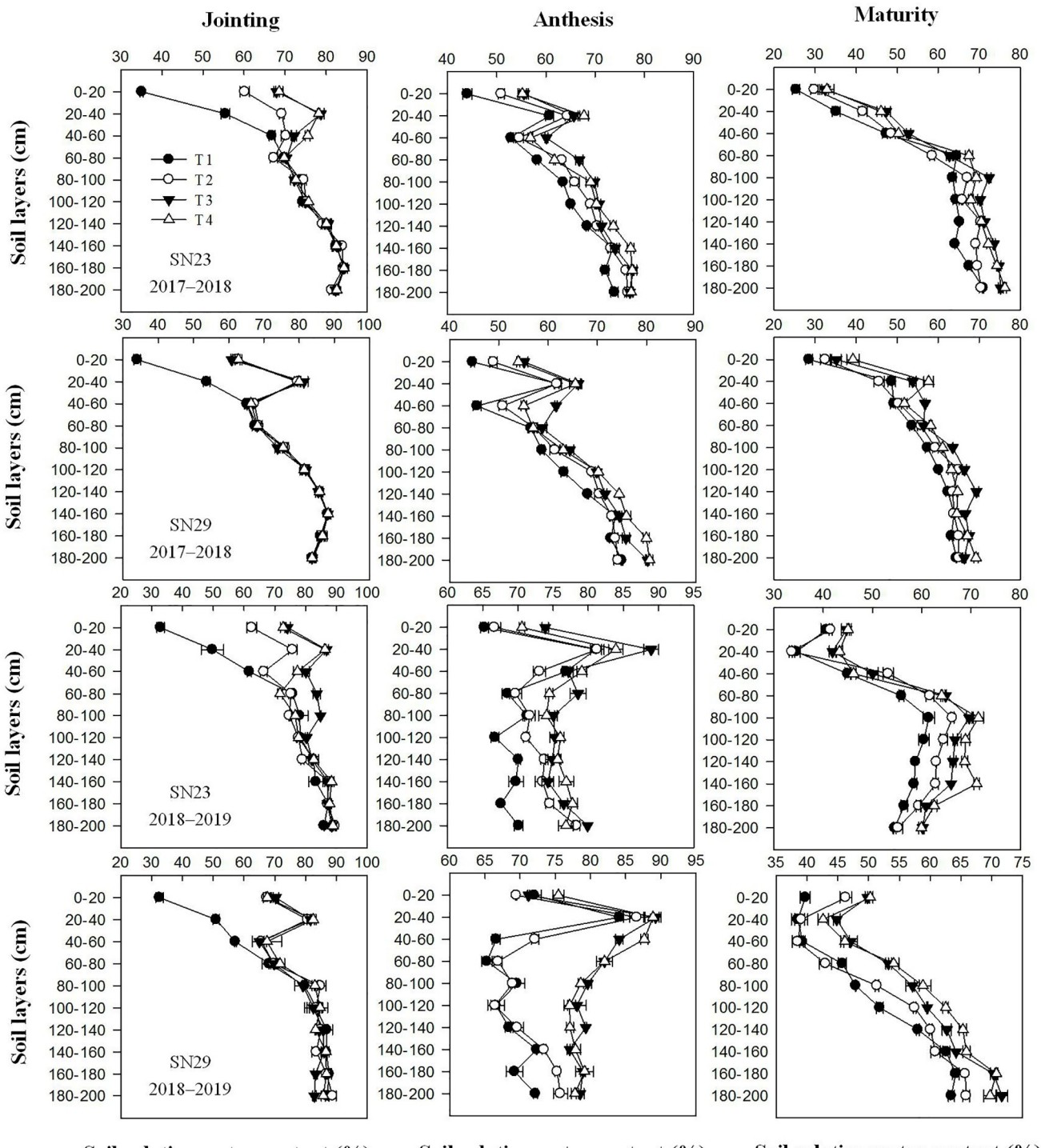

**Fig 1. Soil relative water content in the 0–200 cm soil layers at different growth stages.** The four treatments from 2017–2019 were as follows: no irrigation at jointing (T1), supplemental irrigation with a wetting layer depth of 10 cm at jointing (T2), supplemental irrigation with a wetting layer depth of 20 cm at jointing (T3), and supplemental irrigation with a wetting layer depth of 30 cm at jointing (T4).

**Table 5. Consumption (mm) of soil water in different growth stages of winter wheat from 2017–2019.**

| Year | Cultivar | Treatment | Sowing to wintering | Wintering to jointing | Jointing to anthesis | Anthesis to maturity | Total water consumption |
|---|---|---|---|---|---|---|---|
| 2017–2018 | SN23 | T1 | 10.1a | 63.6a | 56.1b | 170.4a | 300.2c |
| | | T2 | 10.1a | 63.6a | 58.5b | 178.2a | 310.5b |
| | | T3 | 10.1a | 63.6a | 64.4b | 167.8a | 305.9bc |
| | | T4 | 10.1a | 63.6a | 85.3a | 166.9a | 326.7a |
| | SN29 | T1 | 8.6a | 51.1a | 79.9b | 174.6a | 314.1c |
| | | T2 | 8.6a | 51.1a | 91.9b | 178.8a | 330.3b |
| | | T3 | 8.6a | 51.1a | 87.3b | 180.4a | 327.4b |
| | | T4 | 8.6a | 51.1a | 108.2a | 182.5a | 351.0a |
| 2018–2019 | SN23 | T1 | 24.8a | 98.9a | 100.0b | 145.7b | 369.3c |
| | | T2 | 24.8a | 98.9a | 82.8c | 164.6a | 371.0c |
| | | T3 | 24.8a | 98.9a | 90.1bc | 160.7a | 374.5b |
| | | T4 | 24.8a | 98.9a | 116.0a | 145.3b | 387.5a |
| | SN29 | T1 | 22.7a | 97.7a | 100.4b | 161.1b | 381.9b |
| | | T2 | 22.7a | 97.7a | 98.1b | 178.9a | 397.5a |
| | | T3 | 22.7a | 97.7a | 91.0b | 171.3ab | 382.8b |
| | | T4 | 22.7a | 97.7a | 117.7a | 158.5b | 399.6a |

SN23 and SN29 indicate Shannong 23 and 29, respectively. For the same year and within a given cultivar, the different letters in a given column are significantly different at P < 0.05 based on an LSD test.

the T1 treatment. These results indicated that moderate SI at jointing was more beneficial for increasing the number of spikes in the large-spike cultivars than in the multispike cultivars.

## Gas exchange characteristics of different tillers after jointing

The Pn, Tr, and WUE$_{leaf}$ of the apical leaves of different tillers measured at 0, 7, 14, 21, and 28 days after the jointing stage are shown in Figs 3–5, respectively. After irrigation at jointing, no significant differences were found between the T3 and T4 treatments, but the apical leaf Pn of the main stem (O) and tillers I and IV in the T3 and T4 treatments were significantly greater than those in the T2 treatment. The apical leaf Tr of the O and tillers I and IV were significantly higher in the T4 treatment than in the T3 treatment. However, the trend of the WUE$_{leaf}$ was opposite to the trends of the Pn and Tr. The results indicated that excessive irrigation did not significantly improve the photosynthetic capacity of tillers but did significantly increase the leaf Tr, which substantially reduced the WUE$_{leaf}$.

## Grain yield and WUE

The grain yield, yield components, and WUE are listed in Table 6. Compared with the average grain yield in the T1 treatment, that in the T2, T3 and T4 treatments showed increased average grain yield, by 15.60%, 29.68%, and 29.75% in SN23 and by 4.69%, 10.48% and 9.14% in SN29. The increase in grain yield of the large-spike cultivar (SN23) was greater than that of the multi-spike cultivar (SN29). No significant difference in the grain yield was found between the T3 and T4 treatments, which indicated that excessive irrigation at the jointing stage did not significantly increase yield. The WUE of the T3 treatment was the highest in both cultivars. Compared with the T3 treatment, there was no significant difference in the water consumption in the T2 treatment, which also had a lower grain yield. The T4 treatment had no significant

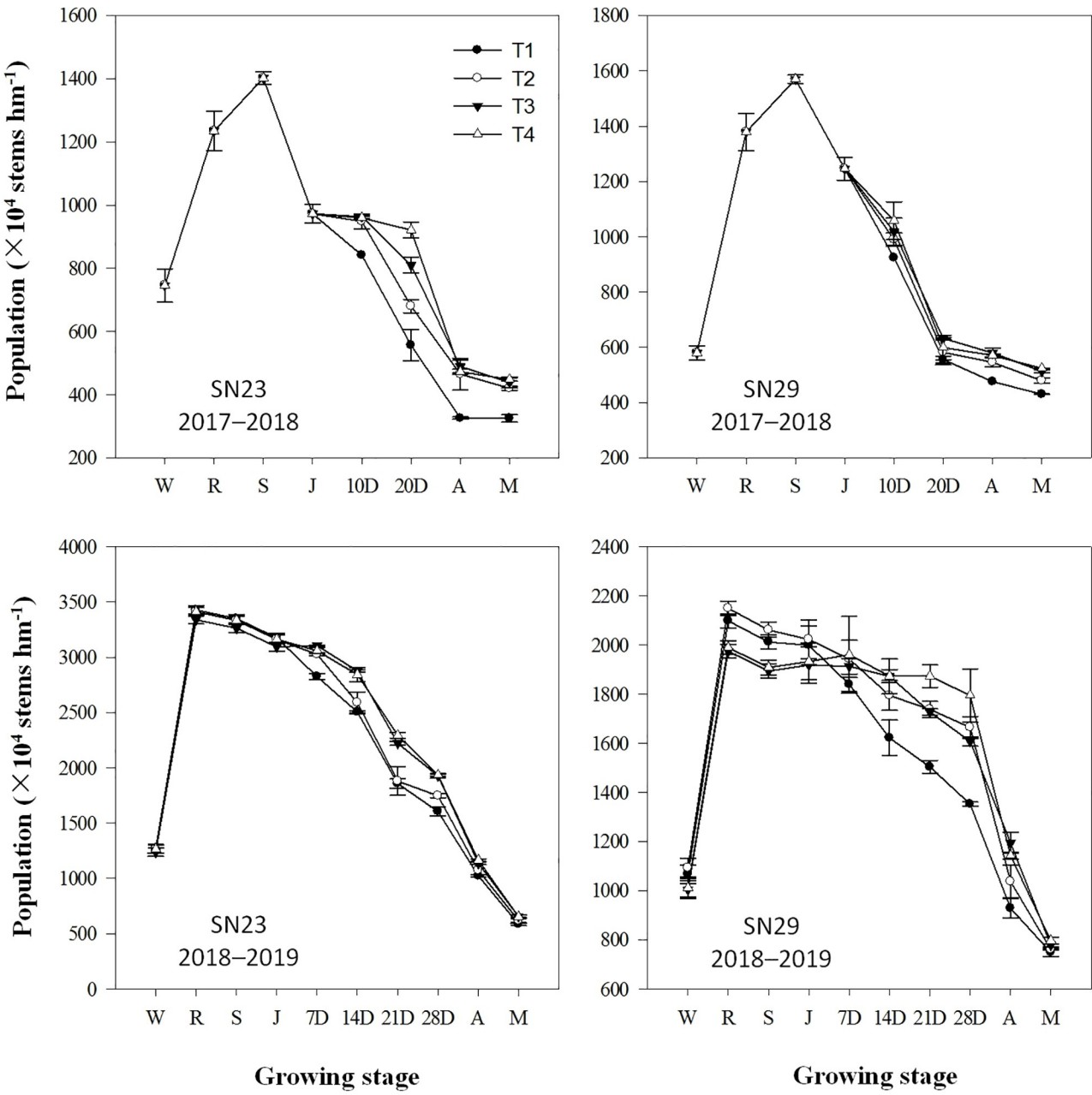

**Fig 2. Population dynamics of winter wheat at different growth stages.** The four treatments from 2017–2019 were as follows: no irrigation at jointing (T1), supplemental irrigation with a wetting layer depth of 10 cm at jointing (T2), supplemental irrigation with a wetting layer depth of 20 cm at jointing (T3), and supplemental irrigation with a wetting layer depth of 30 cm at jointing (T4). W, R, S, J, A, and M indicate the growth stages of wintering, re-greening, standing, jointing, anthesis, and maturity, respectively. D indicates the day after the jointing stage.

difference in the grain yield but had higher water consumption and lower WUE than those of the T3 treatment. These results indicated that regardless of the type of wheat (i.e., large- or multispike), the optimum wetting layer for irrigation at jointing was the 0–20 cm soil layer, which resulted in both high grain yield and WUE.

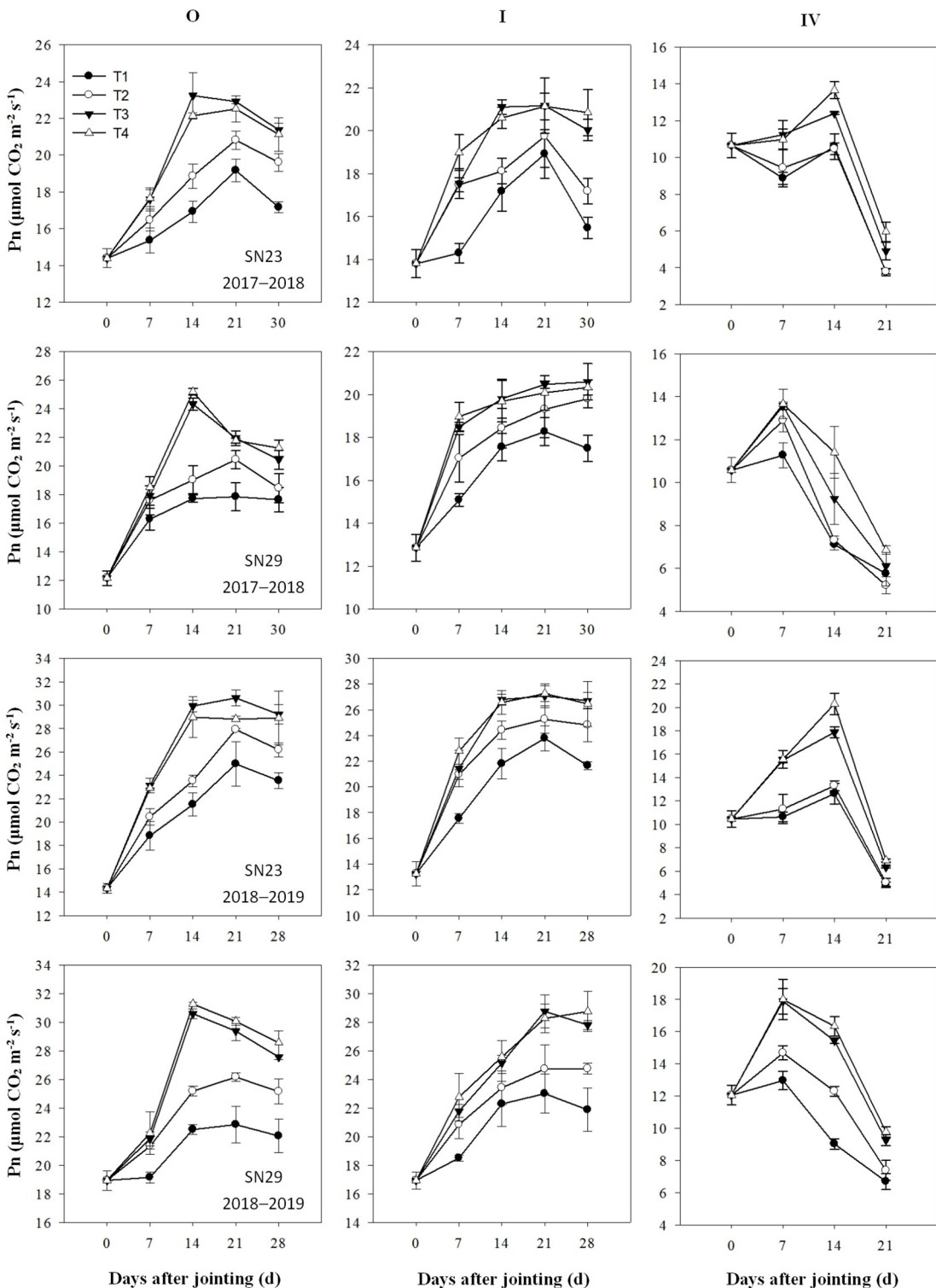

**Fig 3. Net photosynthetic rates (Pn) in tillers after jointing.** The four treatments from 2017–2019 were as follows: no irrigation at jointing (T1), supplemental irrigation with a wetting layer depth of 10 cm at jointing (T2), supplemental irrigation with a wetting layer depth of 20 cm at jointing (T3), and supplemental irrigation with a wetting layer depth of 30 cm at jointing (T4). "O", "I", and "IV" tagged at wintering indicate the main stem and the first and fourth primary tillers, respectively.

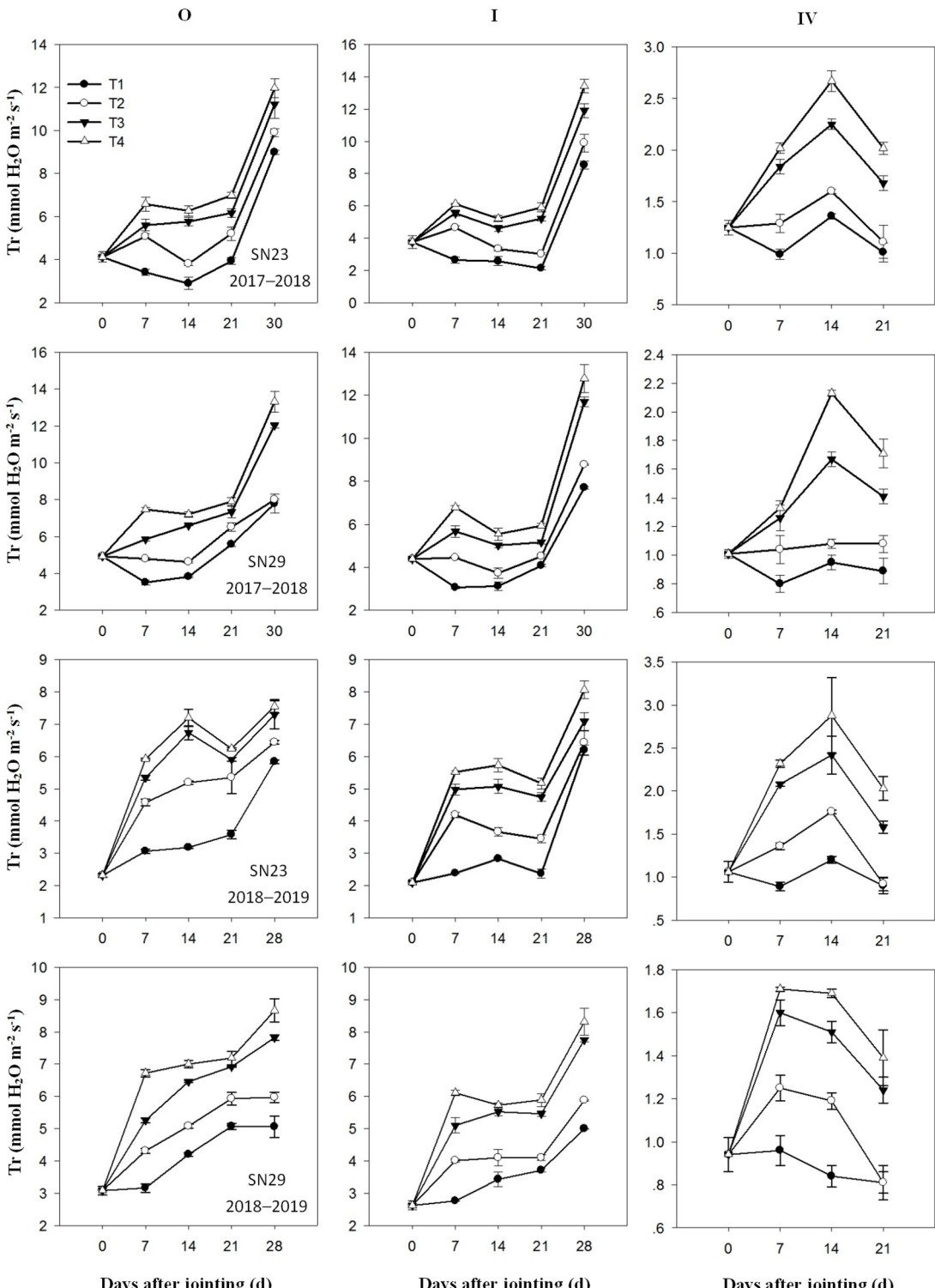

**Fig 4. Transpiration rates (Tr) in tillers after jointing.** The four treatments from 2017–2019 were as follows: no irrigation at jointing (T1), supplemental irrigation with a wetting layer depth of 10 cm at jointing (T2), supplemental irrigation with a wetting layer depth of 20 cm at jointing (T3), and supplemental irrigation with a wetting layer depth of 30 cm at jointing (T4). "O", "I", and "IV" tagged at wintering indicate the main stem and the first and fourth primary tillers, respectively.

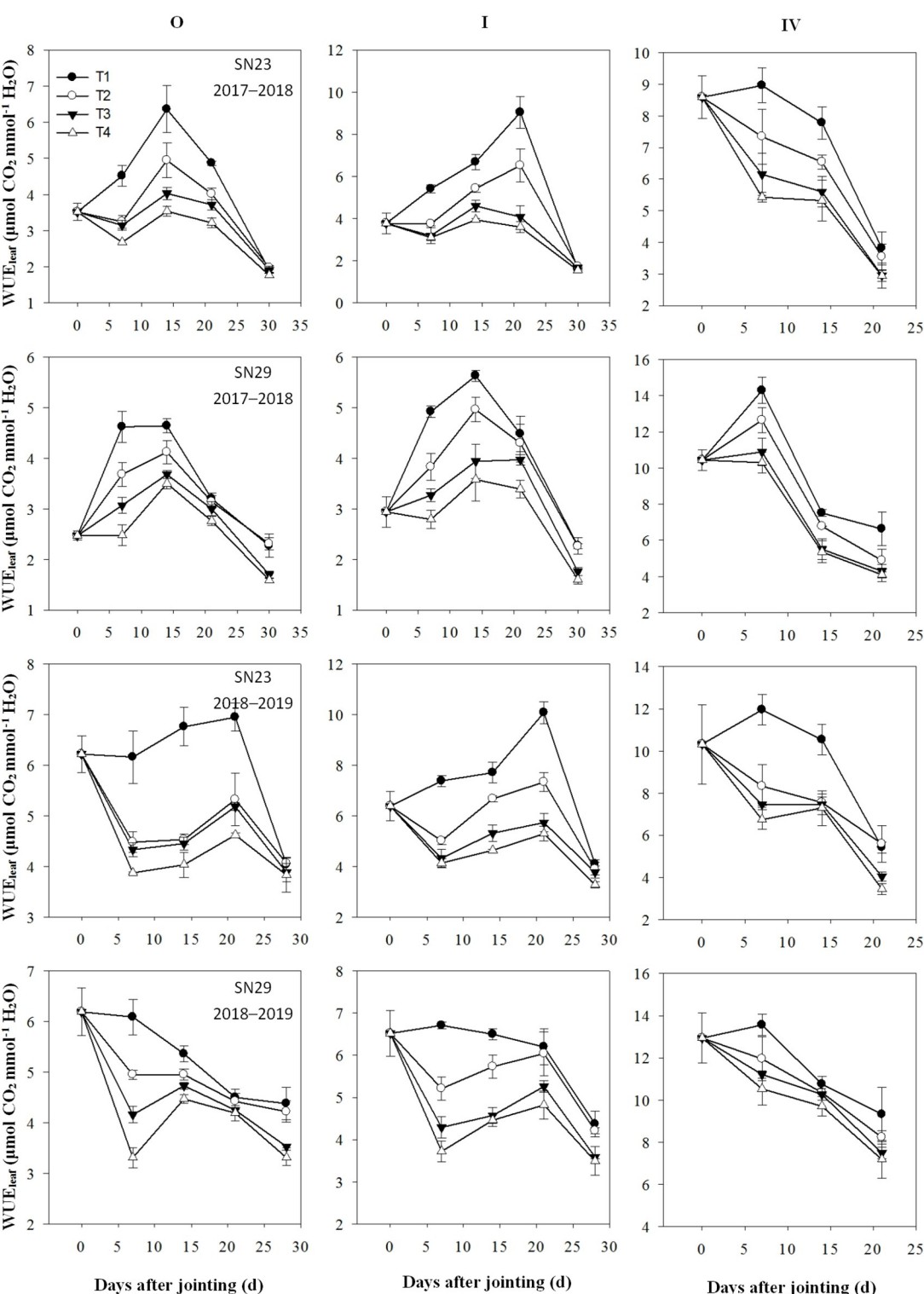

**Fig 5. Instantaneous water-use efficiency (WUE$_{leaf}$) in tillers after jointing.** The four treatments from 2017–2019 were as follows: no irrigation at jointing (T1), supplemental irrigation with a wetting layer depth of 10 cm at jointing (T2), supplemental irrigation with a wetting layer depth of 20 cm at jointing (T3), and supplemental irrigation with a wetting layer depth of 30 cm at jointing (T4). "O", "I", and "IV" tagged at wintering indicate the main stem and the first and fourth primary tillers, respectively.

**Table 6. Grain yield, yield components, and WUE of winter wheat in 2017–2019.**

| Year | Cultivar | Treatment | Spike number (×10⁴·ha⁻¹) | Grains per number | 1000-grain weight (g) | Yield (kg·ha⁻¹) | WUE (kg·ha⁻¹·mm⁻¹) |
|---|---|---|---|---|---|---|---|
| 2017–2018 | SN23 | T1 | 325.24c | 38.00c | 49.13a | 6052.03c | 20.17d |
| | | T2 | 420.62b | 37.86c | 47.57b | 7213.70b | 23.24c |
| | | T3 | 439.21a | 45.56a | 43.17d | 8531.04a | 27.89a |
| | | T4 | 448.60a | 41.88b | 45.83c | 8501.15a | 26.03b |
| | SN29 | T1 | 430.56c | 28.80a | 52.27a | 6584.37c | 20.96b |
| | | T2 | 479.96b | 29.18a | 52.88a | 6915.37b | 20.94b |
| | | T3 | 513.20a | 29.00a | 50.45b | 7242.37a | 22.13a |
| | | T4 | 524.08a | 29.20a | 49.35c | 7260.70a | 20.69b |
| 2018–2019 | SN23 | T1 | 587.61c | 37.14b | 41.30a | 8573.28c | 25.03d |
| | | T2 | 620.20b | 37.59b | 41.68a | 9602.38b | 25.88b |
| | | T3 | 656.17a | 38.23b | 42.41a | 10151.05a | 27.11a |
| | | T4 | 654.16a | 39.98a | 41.97a | 10204.08a | 25.43c |
| | SN29 | T1 | 748.94b | 29.69b | 47.82a | 10030.05c | 26.26c |
| | | T2 | 761.40b | 33.09a | 46.02b | 10466.99b | 26.34c |
| | | T3 | 778.14ab | 33.99a | 45.38bc | 11129.76a | 29.08a |
| | | T4 | 797.24a | 33.42a | 44.82c | 10832.79ab | 27.11b |

SN23 and SN29 indicate Shannong 23 and 29, respectively. For the same year and within a given cultivar, the different letters in a given column are significantly different at P < 0.05 based on an LSD test.

## Discussion

### Relation of the depth of the wetting layers for irrigation at jointing to population dynamics

Under irrigation conditions, the highest moisture treatment was accompanied by the highest number of tillers [17]. Wheat that received irrigation four times had 15%–20% more productive tillers than wheat that received irrigation two or three times, and the yield increased by 18%–40% [28]. Feng et al. (2017) also illustrated that the spike number of plants irrigated three times is greater than that of plants irrigated two times [29]. However, excessive tillering was not conducive to an increase in the number of ears. Irrigation with 75 mm at the end of the tillering period greatly increased the maximum tiller number (by 90%), but the number of productive tillers was reduced to 37% [15]. In this study, the relative water content in the 0–80 cm soil layer increased with the increasing depth of the wetting layers used for irrigation at the jointing stage (Fig 1). There were no significant differences in the relative water content in the 0–20 cm soil layer between the T3 and T4 treatments, while the water content in the T4 treatment was significantly higher than that in the T3 treatment in the 0–80 cm soil layer on the third day after irrigation at the jointing stage (Fig 1). Although the T4 treatment reduced the rate of tiller mortality significantly from jointing to anthesis compared to that in the T3 treatment, there was no significant difference in the spike number at maturity between the T3 and T4 treatments (Fig 2). These results indicated that using the 0–20 cm soil layer as the wetting layer for irrigation at the jointing stage made the relative water content in the 0–80 cm soil layer reach 70.0%–70.8%, which could support a reasonable wheat population while using less irrigation water (Figs 1 and 2). The excessive water supply that occurred with an increased wetting layer depth at the jointing stage could delay the death of ineffective tillers, but it could not improve the spike number (Fig 2).

### Effects of irrigation at jointing on grain yield and WUE

The maximum yield of winter wheat is usually obtained with optimal irrigation, and reducing crop water use by 16% from the full water supply does not affect grain production [30]. Wang and Yu (2008) reported that wheat yield increased with increased irrigation amount when the total water supply was less than 120 mm at jointing and anthesis [31]. When the water supply increased to 120 mm at the jointing stage, the number of spikes and the grain yield increased by 9.1% and 25%, respectively, compared with those given a 60 mm water supply [32,33]. In this study, compared with those in the T3 treatment, the T2 treatment had no significant difference in the kernel number per spike but considerable decreases in the spike number, grain yield, and WUE (Table 6). The T4 treatment achieved the same level of spike number, kernel number per spike, and grain yield but had a significantly reduced WUE. The T3 and T4 treatments reduced the rate of tiller mortality and increased the leaf Pn of the tillers (Figs 2 and 3), which increased the number of productive tillers compared with that in the T2 treatment. In the T4 treatment, the leaf Tr of each tiller significantly increased (Fig 4) and the $WUE_{leaf}$ of the tillers significantly decreased compared with those in the T3 treatment. This demonstrates that excessive water supply at jointing leads to an increase in inefficient water consumption and a decrease in WUE.

### Differences between large- and multispike cultivars under different wetting layer treatments

Obvious differences in spike formation rates were found between the large- and multispike cultivars, with the multispike cultivar having twice the number of productive tillers as that of the large-spike cultivar [34]. In this study, when the relative water content in the 0–20 cm soil layers at jointing was 33.3–33.9%, the spike number increased by 17.64–24.84% in SN23 and 6.57–14.08% in SN29 when irrigation was applied at jointing and the soil water content in the 0–20 cm profile was increased to 100% of field capacity (Figs 1–2 and Table 6). These results indicated that the proportion and number of tillers modified into productive tillers in SN23 were more sensitive to the soil moisture condition at the jointing stage than those in SN29. Other data also support this conclusion. Previous studies have shown that compared with the spike numbers in the rainfed large- and multispike cultivars, those in cultivars that were irrigated twice increased by 87.0% and 57.8%, respectively [35]. There was no significant difference among the T2, T3 and T4 treatments in the kernels per spike in SN29 for either year, while the kernels per spike in the T3 and T4 treatments were higher than that in the T2 and T1 treatments in SN23. The grain yield of the T3 treatment increased by 29.68% in SN23 and 10.48% in SN29 compared with that of the T1 treatment (Table 6). Therefore, the increase in spike numbers and grain numbers per spike was the reason for the greater increase in the grain yield of the large-spike cultivar than that of the multispike cultivar.

### Conclusion

The proportion and number of tillers altered into productive tillers in the large-spike cultivar (SN23) were more sensitive to the soil moisture condition at the jointing stage than those in the multispike cultivar (SN29). Moderate irrigation at jointing reduced the tiller mortality, improved the leaf Pn of the tillers, and increased the spike number and grain yield. However, excessive irrigation significantly increased the leaf Tr of the tillers and significantly reduced the $WUE_{leaf}$ of the tillers and the WUE of both cultivars. Irrigation at the jointing stage brought the soil water content in the 0–20 cm layer to 100% of field capacity

and was the most suitable SI regime for both the large- and multispike cultivars in the North Chain Plain.

## Author Contributions

**Conceptualization:** Yunqiu Shang, Xiang Lin, Dong Wang.

**Data curation:** Yunqiu Shang, Xiang Lin, Ping Li.

**Formal analysis:** Yunqiu Shang, Ping Li, Dong Wang.

**Investigation:** Yunqiu Shang, Ping Li, Keyi Lei, Sen Wang, Xinhui Hu, Panpan Zhao.

**Methodology:** Xiang Lin, Shubo Gu, Dong Wang.

**Project administration:** Dong Wang.

**Writing – original draft:** Yunqiu Shang.

**Writing – review & editing:** Dong Wang.

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
