## [Decision Letter · Decision Letter 0]

5 Jan 2020

PONE-D-19-34837

Effects of supplemental irrigation on population dynamics, grain yield, and water use efficiency of two different spike-type wheat cultivars at jointing stage

PLOS ONE

Dear Mr Shang,

Thank you for submitting your manuscript to PLOS ONE. After careful consideration, we feel that it has merit but does not fully meet PLOS ONE’s publication criteria as it currently stands. Therefore, we invite you to submit a revised version of the manuscript that addresses the points raised during the review process.

We would appreciate receiving your revised manuscript by Feb 19 2020 11:59PM. To enhance the reproducibility of your results, we recommend that if applicable you deposit your laboratory protocols in protocols.io, where a protocol can be assigned its own identifier (DOI) such that it can be cited independently in the future. For instructions see: http://journals.plos.org/plosone/s/submission-guidelines#loc-laboratory-protocols

We look forward to receiving your revised manuscript.

Kind regards,

Aimin Zhang, Ph.D.

Academic Editor

PLOS ONE

Journal Requirements:

This work was financially supported by the National Special Support Program for High-level Talents, the Special Fund for Agro-scientific Research in the Public Interest (201503130), the Major Scientific and Technological Innovation Project of Shandong Province (2019JZZY010716), and the Taishan Industry Leader Talent Project of Shandong.

We note that one or more of the authors are employed by a commercial company: Zibo HeFeng Seed Technology co., ltd.

Reviewers' comments:

Reviewer's Responses to Questions

**Comments to the Author**

1. Is the manuscript technically sound, and do the data support the conclusions?

Reviewer #1: Yes

Reviewer #2: Yes

2. Has the statistical analysis been performed appropriately and rigorously? 

Reviewer #1: Yes

Reviewer #2: Yes

3. Have the authors made all data underlying the findings in their manuscript fully available?

Reviewer #1: Yes

Reviewer #2: Yes

4. Is the manuscript presented in an intelligible fashion and written in standard English?

Reviewer #1: No

Reviewer #2: Yes

5. Review Comments to the Author

Reviewer #1: In this study, Shang et al. investigated the effects of the different irrigation regimes on the tiller numbers, grain yield, and WUE in the two wheat cultivars (Shangnong 23 and Shannong 29) during the jointing stage. The results was interesting and could be beneficial for the practical field work in northern part of China, where the water deficit is one of the major issues limiting the crop yield. However, the manuscript was drafted poorly, with too many format/grammar mistakes. Please ask a native English-speaker to polish this paper. Here I would like point out two major issues: 1. The quality of all the figures are really bad, please use the higher-resolution ones; 2. The discussion is too short, just another simple version of the result part. Please rephrase this part; 3. Most of the references are from China, please try to read some English papers.

Reviewer #2: Improving the grain yield and water use efficiency (WUE) is very important for wheat production, the authors systematically investigated the effects of supplemental irrigation at jointing stage on grain yield and WUE with two wheat cultivars, and also characterized the population dynamics, Pn and Tr of apical leaves. The results provide detailed measurements and comprehensive view of the dynamic regulation of productive tillers.

My suggestions are as follows:

1. According to LSD test performed, the significance should be labeled properly.

2. The apical leaf Pn of main stem, I, and IV tillers were measured, how about other tillers?

3. The differences in spike number, grain weight and other traits between the two cultivars under different plan wetting layers should be discussed.

4. Conclusions should be more unequivocal. In abstract, “All in all, the irrigation brought soil water in the 0-20 cm profile to 100% field capacity is the most suitable supplemental irrigation …”, the related support data for this conclusion should more clear and correspondingly showed in the results parts.

5. There are some typographical errors or inaccurate words, such as page 3 line 72..., page 4 line 90, page 5 line 96.

6. The table 4 can be deleted.

7. The references need to be unified, for example, line 346,347, 368, and 406.

6. PLOS authors have the option to publish the peer review history of their article (what does this mean?). If published, this will include your full peer review and any attached files.

Reviewer #1: Yes: Wenshan Guo

Reviewer #2: No

---

## [Author Response · Author response to Decision Letter 0]

17 Feb 2020

Dear Prof. Zhang,

Thank you for your kind consideration,

We have carefully revised and corrected the Manuscript ID PONE-D-19-34837 entitled “Effects of supplemental irrigation on population dynamics, grain yield, and water use efficiency of two different spike-type wheat cultivars at jointing stage” which we submitted to PLOS ONE, according to your comments and those of the reviewers. 

Funding Statement

The commercial company of Zibo HeFeng Seed Technology co., ltd. just provided support in the form of salaries for authors [Dong Wang], but did not have any additional role in the study design, data collection and analysis, decision to publish, or preparation of the manuscript. So we deleted the commercial company in the new revision. The Funding Statement in the new revised manuscript was described as follows:

This work was financially supported by the Major Scientific and Technological Innovation Project of Shandong Province (2019JZZY010716), the Taishan Industry Leader Talent Project of Shandong, and the Special Fund for Agro-scientific Research in the Public Interest (201503130).

Competing Interests Statement

There was no competing interests exists in this paper.

Responses to the reviewers' comments are presented point by point below.

Responses to Reviewer #1

Following the comments’ order:

The new revision had been edited by the American Journal Experts (AJE) which was recommended by the PLOS editorial team. The changes indicated by red font, can be seen in the new version directly. 

According to the Reviewer’s suggest, we had replaced the figures by the higher-resolution ones in the new revision (Fig. 1–5 in the new revision).

Discussion

We had rephrased the discussion and also added one paragraph at the end of the Discussion to clarify the differences between large- and multispike cultivars under different wetting layer treatments (Line 269–324 in the new revision).

Responses to Reviewer #2

Following the comments’ order:

Question: The apical leaf Pn of main stem, I, and IV tillers were measured, how about other tillers?

Answer: The spike number per plant is 1.6–2.5 stem in the large-spike cultivar SN23, and 2.9–3.5 stem in the multispike cultivar SN29. In order to reduce the workload, only representative tillers were selected for the measurement. Therefore, we only measured the main stem, tillers I and IV, where the tiller I represent as the productive tiller and the tiller IV represent as the inefficient tiller.

The new revision had been edited by the American Journal Experts (AJE) which was recommended by the PLOS editorial team. The changes indicated by red font, can be seen in the new version directly. 

Abstract 

Line 49: The last sentence in the Abstract had been rewritten as “Irrigation at the jointing stage brought the soil water content in the 0–20 cm profile to 100% of field capacity, making it the most suitable supplemental irrigation regime for both the large- and multispike cultivars in the North China Plain” (Line 30 in the new revision).

Discussion

According to the Reviewer’s suggest, one passage was added at the end of the Discussion to clarify the differences between large- and multispike cultivars under different wetting layer treatments (Line 307–324 in the new revision).

Tables

Table 4: According to the Reviewer’s suggest, we deleted it and updated the order of Tables.

Table 6: The significance according to the LSD test performed had been labeled (Table 5 in the new revision).

References

Line 346: “272–273 (2019) 12–19” was changed to “2019; 272–273: 12–19” (Line 349 in the new revision).

Line 347: “Ha, GJ” was changed to “Han GJ” (Line 350 in the new revision).

Line 368: “Plant Population Effects on Growth and Yield in Water-Seeded Rice” was changed to “Plant population effects on growth and yield in water-seeded rice” (Line 371 in the new revision).

Based on our checking carefully, other revisions also were made:

The changes indicated by red font, can be seen in the new version directly.

Title 

Line 1–2: the title was rephrased by “Effects of supplemental irrigation at the jointing stage on population dynamics, grain yield, and water-use efficiency of two different spike-type wheat cultivars” (Line 1–2 in the new revision).

Introduction

Line 70–94: The part of Introduction had been rephrased in the new revision (Line 51–71 in the new revision).

Measurements 

Soil water content and water consumption by winter wheat

Line 171–176: The calculation of the total water consumption of winter wheat has been rewritten as “The total water consumption of winter wheat was calculated using the soil water balance equation [27]: ET = S + I + P – R – D + CR, where ET (mm) is the total water consumption, S (mm) is the water consumption from the 0–200 cm soil layer during the growing season, I (mm) and P (mm) are the irrigation and precipitation, respectively, R (mm) is the runoff, D (mm) is the drainage from the root zone, and CR (mm) is the capillary rise to the root zone. Three factors in this equation (R, D and CR) can be ignored in this experimental site” (Line 142–147 in the new revision).

Population dynamics of winter wheat

Line 178–183: The marking and sampling method of tillers and the method of population survey have been rewritten as “Tillers were marked from the appearance of the first tiller in wheat. The newly initiated tillers of each plant were checked and tagged every 5 days. After jointing, the tagged plants were selected and separated according to the tiller positions for measuring. In this paper, the main stem is represented by O, and the primary tillers growing from the true leaf axillary of O are represented by I, II, III, IV, etc. Conversely, I-p, I-1, I-2, etc. are used for the secondary tillers growing from the axillary of the true leaf of the primary tillers [28]. The number of tillers (stems) per square meter was investigated at wintering, re-greening, standing, jointing, anthesis and maturity. At jointing, the number of tillers (stems) was investigated from 0 days after jointing to anthesis and it was checked once every 7 days” (Line 149–157 in the new revision).

References 

Eight references were deleted and six new references were added in Line 389–398, Line 409–410, and Line 419–423 in the new revision, respectively.

---

## [Decision Letter · Decision Letter 1]

3 Mar 2020

Effects of supplemental irrigation at the jointing stage on population dynamics, grain yield, and water-use efficiency of two different spike-type wheat cultivars

PONE-D-19-34837R1

Dear Dr. Shang,

We are pleased to inform you that your manuscript has been judged scientifically suitable for publication and will be formally accepted for publication once it complies with all outstanding technical requirements.

With kind regards,

Aimin Zhang, Ph.D.

Academic Editor

PLOS ONE

Additional Editor Comments (optional):

Reviewers' comments:

Reviewer's Responses to Questions

**Comments to the Author**

1. If the authors have adequately addressed your comments raised in a previous round of review and you feel that this manuscript is now acceptable for publication, you may indicate that here to bypass the “Comments to the Author” section, enter your conflict of interest statement in the “Confidential to Editor” section, and submit your "Accept" recommendation.

Reviewer #1: All comments have been addressed

Reviewer #2: All comments have been addressed

2. Is the manuscript technically sound, and do the data support the conclusions?

Reviewer #1: Yes

Reviewer #2: Yes

3. Has the statistical analysis been performed appropriately and rigorously? 

Reviewer #1: Yes

Reviewer #2: Yes

4. Have the authors made all data underlying the findings in their manuscript fully available?

Reviewer #1: Yes

Reviewer #2: Yes

5. Is the manuscript presented in an intelligible fashion and written in standard English?

Reviewer #1: Yes

Reviewer #2: Yes

6. Review Comments to the Author

Reviewer #1: (No Response)

Reviewer #2: the revised version of the manuscript is greatly improved and all of the comments raised from reviewers have been addressed.

7. PLOS authors have the option to publish the peer review history of their article (what does this mean?). If published, this will include your full peer review and any attached files.

Reviewer #1: No

Reviewer #2: No

---

## [Editor Report · Acceptance letter]

18 Mar 2020

PONE-D-19-34837R1 

Effects of supplemental irrigation at the jointing stage on population dynamics, grain yield, and water-use efficiency of two different spike-type wheat cultivars 

Dear Dr. Shang:

I am pleased to inform you that your manuscript has been deemed suitable for publication in PLOS ONE. Congratulations! Your manuscript is now with our production department. 

With kind regards,

on behalf of

Prof. Aimin Zhang 

Academic Editor

PLOS ONE